# Effect of Dietary *Chlorella vulgaris* or *Tetradesmus obliquus* on Laying Performance and Intestinal Immune Cell Parameters

**DOI:** 10.3390/ani13101589

**Published:** 2023-05-09

**Authors:** Yoo-Bhin Kim, Jina Park, Yun-Ji Heo, Hyun-Gwan Lee, Byung-Yeon Kwon, Sang Seok Joo, Sung Yong Joo, Myunghoo Kim, Z-Hun Kim, Kyung-Woo Lee

**Affiliations:** 1Department of Animal Science and Technology, Konkuk University, Seoul 05029, Republic of Korea; 2Department of Animal Sceince, College of Natural Resource & Life Sciences, Pusan National University, Miryang 50463, Republic of Korea; 3Life and Industry Convergence Research Institute, Pusan National University, Miryang 50463, Republic of Korea; 4Microbial Research Deparment, Nakdonggang National Institute of Biological Resources (NNIBR), Sangju 37242, Republic of Korea

**Keywords:** *Chlorella vulgaris*, *Tetradesmus obliquus*, laying hens, gut health

## Abstract

**Simple Summary:**

In the agricultural sector, particularly the poultry industry, antibiotics have been used as feed additives to treat and prevent infections or to improve the growth, health, and welfare of chickens. Many nutritional alternative strategies, including natural ingredients, have been investigated for their role in improving gut health without any adverse effect on productivity. The possibility of using microalgae as a new source of nutrients and health additives in animal feed formulations has been increased. Therefore, the aim of this study was to determine the effect of dietary *C. vulgaris* or *T. obliquus* on the laying performance, egg quality, intestinal histology, immune characteristics, antioxidant status, and gut microbiome of the laying hen.

**Abstract:**

A feeding trial was conducted to investigate the effect of dietary supplementation of *Chlorella vulgaris* (CV) or *Tetradesmus obliquus* (TO) on laying performance, egg quality, and gut health indicators of laying hens. A total of 144 Hy-Line Brown laying hens aged 21 weeks were randomly assigned to one of three dietary treatments with eight replicates of six hens. Dietary treatments were as follows: CON, basal diet; CV, basal diet + 5 g *C. vulgaris*/kg of diet; TO, basal diet + 5 g *T. obliquus*/kg of diet. The results showed that diets supplemented with CV or TO had insignificant effects on laying performance, egg quality (i.e., Haugh unit and eggshell strength and thickness), jejunal histology, cecal short-chain fatty acids, and antioxidant/immune markers in ileal mucosa samples of laying hens. Compared with the control group, the egg yolk color score was higher (*p* < 0.05) in laying hens fed on diets containing CV and TO, although the former was a more intense yellow than the latter. Small intestinal lamina propria cells were isolated using flow cytometry to examine the percentages of immune cell subpopulations. Dietary microalgae did not affect B cells or monocytes/macrophages but altered the percentage of CD4+ T cells and CD8− TCR γδ T cells. Collectively, diets supplemented with *C. vulgaris* or *T. obliquus* can improve egg yolk color and would modulate host immune development and competence in laying hens.

## 1. Introduction

Microalgae are photosynthetic unicellular species living in saline or freshwater that photosynthetically convert inorganic nutrients into algae biomass, such as macromolecules (i.e., proteins, carbohydrates, and lipids) and microelements including polyphenols, flavonoids, and carotenoids [1]. Because of their nutritious content, microalgae are a possible new source of nutrients and health additives for animal feed formulations [2]. Indeed, dietary microalgae have been used to replace traditional energy and protein ingredients such as corn and soybean meal [1]. In addition, microalgae have been used as feed additives, which improve gut health due to the presence of natural ingredients with antibacterial, anti-inflammation, and antioxidant properties [3,4].

*Chlorella vulgaris* (*C. vulgaris*), a naturally occurring unicellular green microalga, has been artificially cultivated for industrial and nutritional applications worldwide [5]. Feeding laying hens with Chlorella has a beneficial influence on egg production and quality as well as on intestinal lactic acid-producing bacterial populations [6] and produces a more intensive yellow color of the egg yolk [7]. *Tetradesmus obliquus* (*T. obliquus*) is a green microalga that produces bioactives with high added value [8]. Unfortunately, information regarding the role of dietary *T. obliquus* in the laying hen is scarce. However, as *T. obliquus* is known to produce essential nutrients (amino acids and fatty acids) and pigments, including chlorophylls, lutein, and carotenoids [9,10,11], it might be used as a feed additive with health benefits for laying hens. Therefore, the aim of this study in this communication was to determine the effect of dietary supplementation of *C. vulgaris* or *T. obliquus* on the laying performance, egg quality, intestinal histology, antioxidant status, and immune characteristics (including immune cell subpopulations) of the laying hen.

## 2. Materials and Methods

### 2.1. Birds and Experimental Design

A total of 144 21-wk-old laying hens (Hy-Line Brown) were randomly assigned to one of three dietary groups with eight replicates per group. Two hens were raised in a cage (45 cm × 45 cm × 45 cm) in a windowless, fan-ventilated house, and the adjacent three cages were considered a replicate (*n* = 6 birds/replicate). These hens had been vaccinated against a variety of diseases, including Marek’s disease, Newcastle disease, infectious bronchitis, and infectious bursal disease; however, no vaccines were administered after 12 weeks of age. During an experimental period of 4 weeks, laying hens were fed either corn/soybean meal-based basal diets without addition (CON) or with *C. vulgaris* (CV) or *T. obliquus* (TO) at the concentration of 5 g per kg of diet. All diets contained no enzymes or feed additives (other than CV or TO). The ingredients and composition of the basal diet are shown in Table 1. Both CV and TO were provided by the Nakdonggang National Institute of Biological Resources (Sangju, Republic of Korea), and the analyzed nutritional compositions of microalgae are shown in Table 2. Both CV and TO are newly isolated freshwater green microalgae from the Nakdonggang River in South Korea. As the biochemical (e.g., nutritional) compositions of CV and TO can be altered by growth conditions, their concentrations are within the values reported elsewhere [11,12]. Feed and water were provided ad libitum. A lighting program of 15 h of light and 9 h of dark was used throughout the experimental period. The temperature and relative humidity in the experimental room were kept constant at 21 ± 2 °C and 60%.

### 2.2. Laying Performance and Egg Quality

Feed intake per replicate was recorded and used to calculate the daily feed intake per bird. Egg production and egg weight were recorded daily and used to calculate the egg mass. The feed conversion ratio was calculated as the feed intake/egg mass per replicate. On the last three consecutive days at 4 weeks, six intact eggs per replicate were collected for egg quality assessment. The eggshell color was estimated using a shell color reflectometer (TSS QCR, Technical Services and Supplies, York, UK). The Haugh unit, eggshell strength, eggshell thickness (without shell membrane), and yolk color score were assessed using a digital egg tester (DET-6000, Nabel, Kyoto, Japan).

### 2.3. Samplings

At the end of the experiment, one hen per replicate was euthanized by gradual overdose of carbon dioxide, as recommended by the ethical committee. Immediately after euthanasia, the small intestine and a pair of ceca were sampled. For measurement of histology, an approximately 1 cm long mid-jejunal segment was fixed in 10% neutral buffered formalin for 48 h. The remainder of the jejunum was used for lamina propria (LP) cell subpopulation measurements. An ileal segment and a pair of ceca were used to measure antioxidant/immune markers and short-chain fatty acids.

### 2.4. Jejunal Histology

Histological sections (5 μm thick) were stained with hematoxylin and eosin per standard histological technique. The mucosa was examined by a light microscope (Olympus BX43, Tokyo, Japan) and photographed using a digital camera (eXcope T500, DIXI Science, Daejeon, Republic of Korea). Ten intact well-oriented villus and crypts were counted for villus height and crypt depth. Villus height was measured from the villus tip to the villus bottom, and crypt depth was defined as the distance from the villus bottom to the crypt. The ratio of villus height and crypt depth was then calculated.

### 2.5. Antioxidant and Immune Response Markers in Ileal Mucosa

Ileal mucosal scrapings were collected by removing the mid-ileum, rinsing the tissue with PBS, and scraping the mucosal layer from the underlying connective tissue using a microscope slide. The homogenate was centrifuged at 3000 rpm at 4 °C for 10 min, and the supernatant was collected for subsequent measurements. Ileum mucosa samples were used to measure various biomarkers of oxidative stress, including levels of glutathione peroxidase (GPX), superoxide dismutase (SOD), catalase (CAT), malondialdehyde (MDA), and secretory immunoglobulin A (sIgA). GPX and SOD activities were measured by using EnzyChrom superoxide dismutase assay kit (ESOD-100) and EnzyChrom glutathione peroxidase assay kit (EGPX-100), respectively, as per the manufacturer’s instructions. Both kits were bought from BioAssay Systems (Hayward, CA, USA). CAT was analyzed using an OxiSelect catalase activity assay kit (Cell Biolabs, Inc., San Diego, CA, USA). MDA was measured using an OxiSelect thiobarbituric acid reactive substances (TBARS) assay kit (Cell Biolabs, Inc., San Diego, CA, USA). sIgA was determined using a chicken IgA ELISA Kit (Bethyl Laboratories, Montgomery, TX, USA). Bradford reagent (Sigma-Aldrich, St. Louis, MO, USA) was used to quantify the protein concentration of the ileal mucosa samples.

### 2.6. Gut Lamina Propria Cell Isolation

LP cells were isolated using a slightly modified previous method [13]. Briefly, jejunum tissues were cut into 0.5 cm pieces and washed with phosphate-buffered saline (PBS; Thermo Fisher Scientific, Waltham, MA, USA) containing 1 mM DL-dithiothreitol (DTT; Sigma-Aldrich, St. Louis, MO, USA), 30 mM ethylene-diamine-tetra acetic acid (EDTA; Thermo Fisher Scientific, Waltham, MA, USA), and 10 mM 4-[2-hydroxyethyl]-1-piperazineerhanesulfonic acid (HEPES; Thermo Fisher Scientific, Waltham, WA, USA) at 37 °C for 10 min (predigestion first step). Then, tissue samples were washed again in PBS containing 30 mM EDTA and 10 mM HEPES at 37 °C for 10 min (predigestion second step). After the washing step, tissues were transferred to 5 mL of 10% fetal bovine serum (FBS) containing RPMI 1640 (GenDEPOT, Barker, TX, USA) and inverted for 2 min (neutralization step). Lastly, the tissues were digested in 10% FBS containing RPMI 1640 with 0.5 mg/mL collagenase VIII (Sigma-Aldrich, St. Louis, MO, USA) at 37 °C for 1 h (digestion step). After the digestion step, isolated cells were applied to Percoll (GE Healthcare, Chicago, IL, USA) gradient centrifugation (40% Percoll on the top, 70% Percoll on the bottom).

### 2.7. Flow Cytometry Analysis

Isolated LP cells were analyzed using FACS Canto II (BD, Franklin Lakes, NJ, USA). Dead cells were excluded using Live/Dead fixable dead cell stain (Thermo Fisher Scientific, Waltham, MA, USA). The following anti-chicken antibodies were used for staining: anti-CD3 (CT-3; Southern Biotech, Birmingham, AL, USA), anti-CD4 (CT-4; Southern Biotech, Birmingham, AL, USA), anti-CD8a (CT-8; Southern Biotech, Birmingham, AL, USA), anti-TCR γδ (TCR-1; Southern Biotech, Birmingham, AL, USA), anti-MHC II (2G11; Southern-Biotech, Birmingham, AL, USA), anti-Bu-1 (AV20; Southern Biotech, Birmingham, AL, USA), and anti-Monocyte/Macrophage (KUL01; Southern Biotech, Birmingham, AL, USA). All antibodies were diluted 1:200 in PBS and incubated for 30 min under dark conditions. Then, all samples were fixed using 4% paraformaldehyde (PFA) and stored at 4 °C until analysis. The analysis was conducted by two panels: (1) MHC II, Bu-1, and monocyte/macrophage for B cells and APCs; (2) CD3, CD4, CD8a, and TCR γδ for T cells. For the detailed gating strategy of flow cytometry analysis, refer to Appendix A.

### 2.8. Analysis of Short-Chain Fatty Acids (SCFAs) in Cecal Digesta

Approximately 1 g of cecal digesta was homogenized in 4 mL of ice-cold distilled water. The homogenate was then mixed with 0.05 mL of saturated HgCl_2_, 1.00 mL of 25% H_3_PO_4_, and 0.20 mL of 2% pivalic acid and centrifuged at 1000× *g* at 4 °C for 20 min. One milliliter of supernatant was used to measure the concentration of SCFAs by gas chromatography (6890 Series GC System, HP, Palo Alto, CA, USA), as described by Kim et al. [14].

### 2.9. Statistical Analysis

Three adjacent cages were considered the experimental unit. All data were checked for normality by the Shapiro–Wilk test and homogeneity of variance by Levene’s test. The normally distributed data were analyzed by one-way ANOVA followed by Tukey’s test. Non-normally distributed data were analyzed by the Kruskal–Wallis test, followed by Dunn’s test. As all data had equal variances, Welch’s ANOVA was not utilized. All data except for monocytes/macrophages were normally distributed. All statistical analyses were performed using SAS software, version 9.4 (SAS Institute Inc, Cary, NC, USA). Figures for immune cell subpopulations were generated using Prism 8 software (GraphPad Software v8.0.2, San Diego, CA, USA). The significance level was pre-set at *p* < 0.05, and tendency was declared at *p* < 0.10.

## 3. Results

None of the dietary microalgae affected laying performance, including feed intake, egg production, egg weight, egg mass, and feed conversion ratio (Table 3). Both CV and TO increased (*p* = 0.001) yolk color compared with the control group. However, the indicators of egg freshness (Haugh unit) and eggshell quality (i.e., strength and thickness) were not altered by dietary microalgae (Table 4). Dietary microalgae did not affect (*p* < 0.05) villus height, crypt depth, or their ratios in laying hens (Table 5). The relative percentages of cecal SCFAs are presented in Table 6. Acetate dominated, followed by propionate and butyrate in cecal digesta. The oxidative stress/immune response markers (i.e., SOD, MDA, CAT, GPX, and sIgA) in ileal mucosa were not altered by dietary microalgae (Table 7).

The immune cell subpopulations of lamina propria in laying hens were assessed by flow cytometry through two panels (Appendix A). In panel one, MHCII+ cells were first identified, and then the population of B cells and Monocytes/Macrophages were confirmed using Bu-1 (B cell marker) and Mono/Mac (Monocytes/Macrophage marker). No significant differences in subpopulations of the B cells or monocytes/macrophages caused by dietary microalgae (Figure 1) were noted. In panel two, T cells were mainly analyzed. After checking the total T cells (CD3+ cells), various T cell subsets were compared through CD4 and CD8 combinations and TCR γδ and CD8 combinations. The CD3+ T cells tended to increase in CV- and TO-fed laying hens compared with the control group; however, statistical significance was not found (Figure 2). Dietary CV tended to increase the subpopulation of CD4+ T cells compared with the control and the TO-fed groups; however, statistical significance was not found (*p* = 0.054). However, CD8+ T cells were not altered by dietary microalgae. Among the TCR γδ T cells of laying hens, the percentage of the CD8− TCR γδ T cells was less distributed in the TO group compared with the control group. The CD8+ TCR γδ T cells ranged from 38 to 40% and were not affected by dietary treatments.

## 4. Discussion

It is clear from this study that none of the dietary microalgae affected laying performance, jejunal histology, immune or antioxidant markers in ileal mucosa, or cecal SCFAs in laying hens. In line with our study, Halle et al. [15] reported that dietary supplementation of *C. vulgaris* at the inclusion levels from 2.5 to 7.7 g per kg of diet had no effect on laying performance. Alfaia et al. [16] reported that the incorporation of 10% *C. vulgaris* microalgae in broiler diets was not detrimental to the growth performance of broilers. In contrast to our finding, Zheng et al. [6] found that fermented *C. vulgaris* at 0.1 or 0.2% significantly increased egg production without affecting feed intake, egg weight, or egg mass in laying hens. However, it should be remembered that the laying hens employed by Zheng et al. [6] were aged 80 weeks, and egg production was kept low, ranging from 55.4 to 59.0%. On the other hand, we used 21-week-old laying hens with high efficiency of egg production, which might have been unaffected or less affected by dietary additives. 

It is, however, emphasized that the effect of lack of dietary microalgae on laying performance was not related to the absence of biochemical, functional components. In this study, a change in the color of the yolk was clearly noted upon inclusion of the microalgae. Both CV and TO had pigments, including chlorophyll and carotenoids (Table 2), which were incorporated into the egg yolks, intensifying the yellowness of the yolks upon ingestion. Egg yolk color is known to be influenced mostly by the diet of the hens because they are not able to synthesize pigments for yolk but can store them when obtained from the diet [17]. Herber-Mcneill and Van Elswyk [18] reported that the color shift caused by graded levels of dietary microalgae could presumably be explained by the incorporation of microalgal carotenoids into the yolk, specifically canthaxanthin and β-carotene. It might be pointed out that although both CV and TO increased the yolk color, the former was slightly higher than the latter. Table 2 indicates that CV has higher total chlorophyll but low total carotenoid contents than TO. Thus, which pigments are more effective in intensifying the egg yolk color needs to be addressed.

Microalgae are known to contain natural ingredients with antibacterial, anti-inflammation, and antioxidant properties [3,4], which urged us to investigate gut health. In this study, the effect of dietary microalgae on gut health was monitored by assaying gut histology, cecal SCFAs, antioxidant/immune markers on ileal mucosa, and LP cell subpopulations in the jejunum of laying hens. Except for LP cell subpopulations, dietary microalgae did not affect jejunal histology, cecal SCFAs, or antioxidant/immune markers in ileal mucosa. In contrast to our finding, dietary Spirulina and Chlorella-based algae products at 1.75 g per kg of diet decreased intestinal permeability (assayed by fluorescein isothiocyanate-dextran inoculation) and lowered jejunal crypt depth at 7 days following Eimeria challenge in broiler chickens [19]. The former group also reported that dietary algae did not affect intestinal permeability but lowered crypt depth at 14 days post-hatch in naïve broiler chickens. Thus, it is likely that dietary microalgae might be more effective under challenge conditions (e.g., coccidiosis) or in less developed digestive organs in chickens. Whether the dietary microalgae used in this study would improve gut health indices in harsh conditions, such as high stocking density, disease (e.g., coccidiosis, necrotic enteritis), or high temperature/high humidity environment in laying hens, needs to be addressed. 

In this study, we found that the immune cell subpopulation of LP cells in laying hens was altered by microalgae feeding. LP cells contain various types of immune cells, including innate and adaptive immune cells [20]. Of note, the percentage of CD4+ T cells was the highest in the CV group, while CD8− TCR γδ T cells were lowest in TO-fed laying hens compared with the control group. However, B cells and monocytes/macrophages, and T cell subpopulations (i.e., CD8+ T cells and CD8+ TCR γδ T cells) were not altered by dietary microalgae. Among the T cells, CD4+ T cells play a key role in gut homeostasis. For example, it is known that they stimulate the phagocytosis of antigen-presenting cells (APCs) and regulate other immune cells through cytokine secretion [21]. There is also supporting evidence for the existence of CD4+ T cell subsets, such as Th1, Th2, Th17, and Treg cells [22,23]. To identify the characteristics of laying hen CD4+ T cells and subsets, related gene or protein expression analysis should be applied in future studies. Another major finding of this study is that TCR γδ T cells (CD8−) were lower in the TO feed group compared with the control diet group. There are two major chicken T cells populations, αβ− and γδ− TCR T cells [24]. Unlike other mammals, the ratio of γδ T cells in chickens is high (i.e., blood circulating 20–50%, mucosal tissue up to 50%), and they probably play an important role in chickens’ immune systems, including gut immunity [25]. Diverse functions of chicken γδ T cells have been reported in previous studies, such as the production of interferon gamma (IFN− γ) and cytotoxic activity [26,27]. The cytotoxic function of chicken γδ T cells (especially CD8+ γδ T cells) has been reported in Marek’s disease virus challenge study [28]. Briefly, TCR γδ stimulation by viral infection induced IFN-γ production by γδ T cells in chicken peripheral blood mononuclear cells (PBMCs), and the injection of activated PMBCs reduced viral replication and induced cytokine secretion and cytotoxic activity to regulate viral infection. Fenzl et al. reported that CD8+ γδ T cells had higher cytotoxic activity compared to CD8− γδ T cells [27]. In our finding, there was a significant reduction in CD8− γδ T cells, not CD8+ γδ T cells, in the laying hen small intestine LP according to the results from microalgae TO feed. Although studies related to CD8− γδ T cells are insufficient compared to those of CD8+ γδ T cells, changes in CD8− γδ T cells by dietary feed additives will play an important role in gut immunity (especially cytotoxicity function). In future studies, ex vivo or in vivo experiments should be conducted to identify the characteristics of chicken γδ T cells. It is less likely that there would be inflammation at the assayed gut level as monocytes/macrophages (proinflammatory indicator) and antioxidant/oxidative markers in ileal mucosa would not be altered. Nonetheless, dietary microalgae marginally influenced T cell subpopulations of laying hens, indicating their potential immune-regulatory effect. Similarly, Rim et al. [29] reported that TO supplementation altered the percentage of CD4+ T cells and CD8+ T cells in broiler chickens. It is well documented that T cells are primary mediators of immune effector functions and play the most important role in protecting against gut pathogens in chickens [30]. Thus, dietary microalgae (CV and TO) could potentially enhance immune development and competence in the intestine, facilitating a protective immune response against pathogenic bacteria and parasites in laying hens. 

## 5. Conclusions

It is concluded that dietary microalgae (*C. vulgaris* or *T. obliquus*) is a functional ingredient that improves egg yolk color. However, the performance-related parameters, intestinal histology, and antioxidant/immune markers in ileal mucosa were not affected by the inclusion of the microalgae. Of note, dietary microalgae partially influenced the percentage of T cell subpopulations in LP cells of laying hens. Further research is required to determine whether dietary microalgae would augment host immune competence under pathogen exposure in chickens.

## Figures and Tables

**Figure 1 animals-13-01589-f001:**
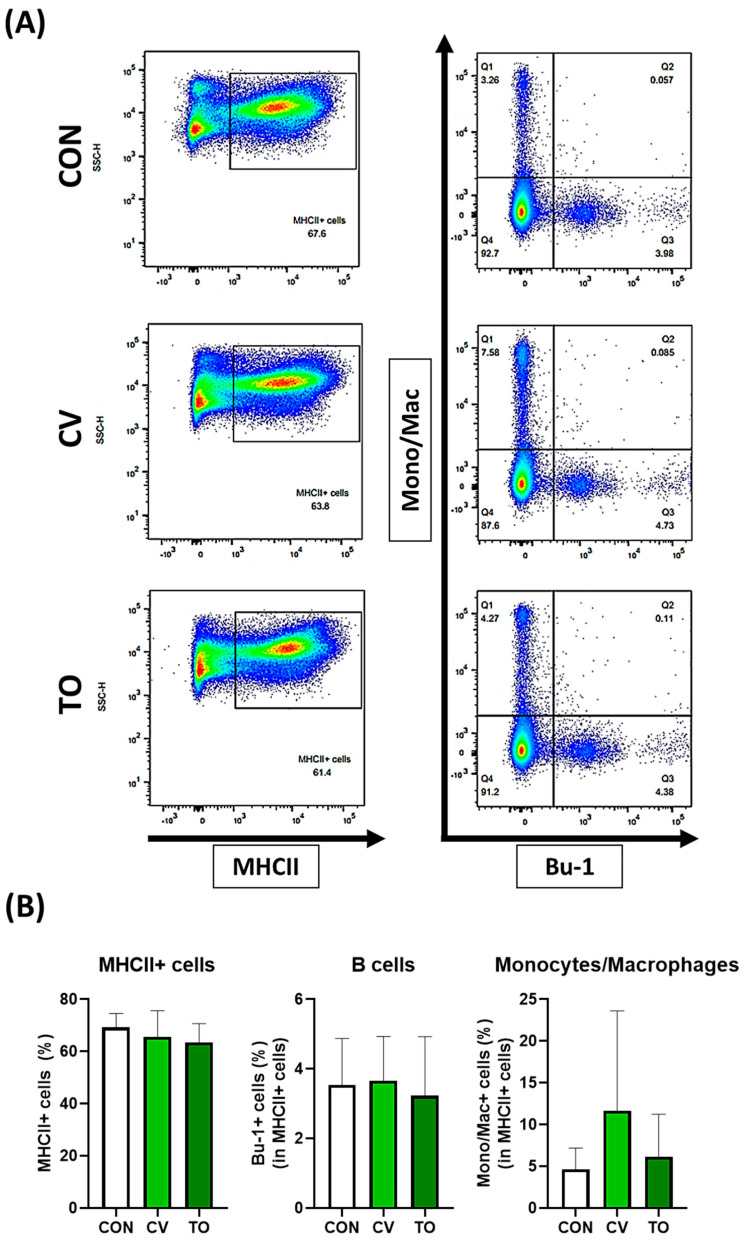
Changes in antigen-processing cells in the small intestine caused by microalgae feeding. Proportions of MHC II+ cells, B cells, and monocyte/macrophages were analyzed by flow cytometry in laying hens. A representative dot plot is shown (**A**), and immune cell distribution is presented (**B**). In the panel (**A**), pseudo color plots indicate cell density.

**Figure 2 animals-13-01589-f002:**
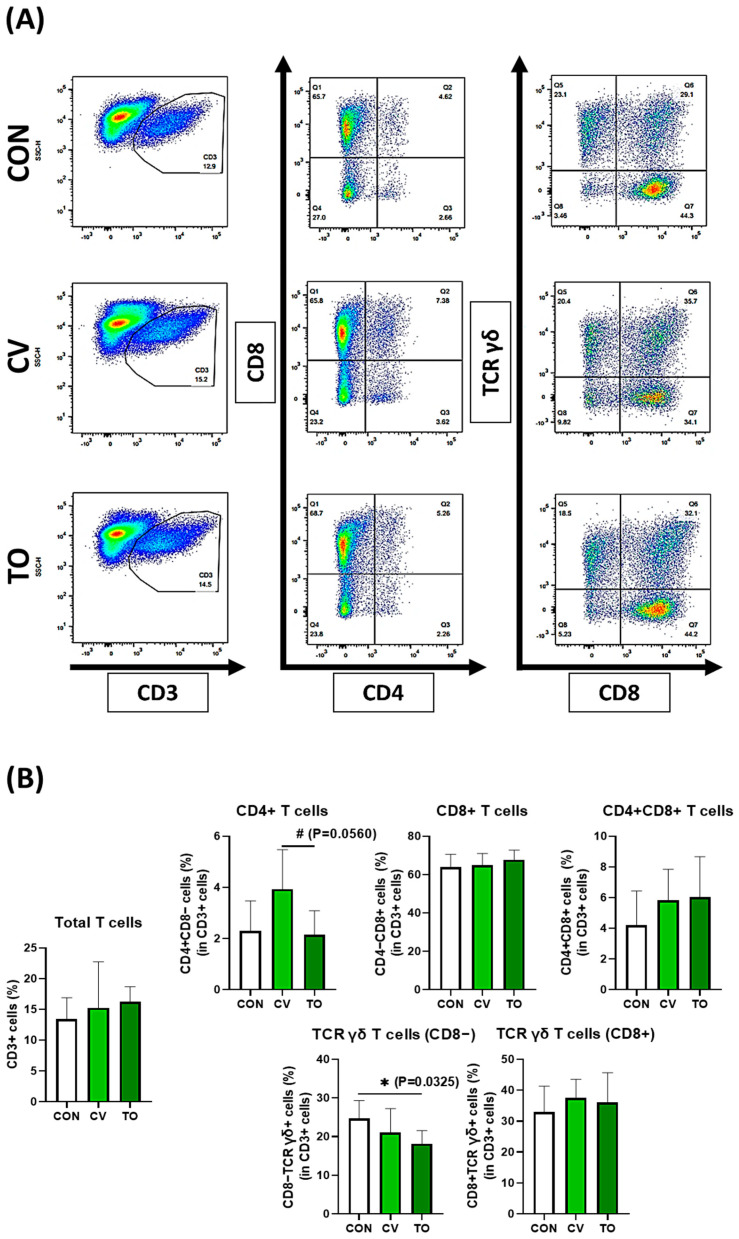
Changes in T cells in the small intestine caused by microalgae feeding. Proportions of total T cells, CD4 T cells, CD8 T cells, and TCR γδ cells were analyzed by flow cytometry in laying hens. A representative dot plot is shown (**A**), and immune cell distribution is presented (**B**). In the panel (**A**), pseudo color plots indicate cell density. The asterisk (*) indicates significance at *p* < 0.05 and the number sign (#) indicates significance at *p* < 0.10.

**Table 1 animals-13-01589-t001:** Ingredient and chemical composition of the basal diet.

Ingredients	g per 100 g of Diet
Corn	56.985
Soybean meal, 44.80% CP	6.909
Corn gluten meal, 67.35% CP	2.271
Wheat	2.126
Sesame oil meal, 41.07% CP	1.594
Beef tallow	0.425
Rapeseed meal, 37.31% CP	2.126
Dried distillers’ grains with solubles, 27.43% CP	14.350
Full-fat soybean, 36.17% CP	0.901
Limestone	10.797
Monocalcium phosphate	0.543
NaCl	0.244
Choline chloride	0.021
Methionine	0.162
Lysine sulfate	0.302
Threonine	0.040
Mineral mix ^1^	0.128
Vitamin mix ^2^	0.074
Total	100.0
Calculated chemical composition, g/100 g	
Nitrogen-corrected apparent metabolizable energy ^3^, kcal/kg	2750
Dry matter ^3^	88.58
Crude protein ^3^	18.00
Crude fat ^3^	3.97
Crude ash ^3^	13.57
Crude fiber ^3^	2.36
Calcium ^3^	4.03
Available phosphorus ^3^	0.40
Lysine ^3^	0.88
Methionine ^3^	0.51
Methionine + Cysteine ^3^	0.80
Threonine ^3^	0.68
Na ^3^	0.15
Cl ^3^	0.18

^1^ Mineral mixture provided the following nutrients per kg of diet: Fe, 70 mg; Cu, 7.5 mg; Zn, 60 mg; Mn, 80 mg; I, 1 mg; Co, 0.1 mg; Se, 0.2 mg. ^2^ Vitamin mixture provided the following nutrients per kg of diet: vitamin A, 20,000 IU; vitamin D_3_, 4600 IU; vitamin E, 40 mg; vitamin K_3_, 4 mg; vitamin B_1_, 3.6 mg; vitamin B_2_, 8 mg; vitamin B_6_, 5.8 mg; vitamin B_12_, 0.04 mg. ^3^ Calculated value.

**Table 2 animals-13-01589-t002:** Analyzed compositions of microalgae used in the current study.

Item, mg/g DCW	*Chlorella vulgaris*	*Tetradesmus obliquus*
Nitrogen-free extract	393.8	352.5
Crude protein	210.0	261.6
Chlorophyll a	84.64	18.49
Chlorophyll b	12.85	4.52
Astaxanthin	0.20	ND
Anteraxanthin	2.06	ND
Lutein	2.85	12.95
Beta-carotene	ND	2.75
Total chlorophyll	97.49	23.01
Total carotenoid	5.11	15.70
C16:0	3.11	30.44
C16:1	22.14	4.80
C16:2	2.10	7.00
C16:3	ND	5.10
C16:4	3.65	5.50
C18:0	8.11	6.30
C18:1	7.18	12.30
C18:2	5.38	13.60
C18:3 α	20.42	15.00

**Table 3 animals-13-01589-t003:** Effect of dietary microalgae on laying performance in laying hens.

Item	CON ^1^	CV	TO	SEM ^2^	*p*-Value
Feed intake, g/bird/day	100.6	99.9	103.1	1.390	0.261
Egg production, %	93.3	89.9	94.1	2.088	0.219
Egg weight, g	56.54	57.07	57.32	0.431	0.442
Egg mass, g/day	52.71	51.33	53.93	1.209	0.335
Feed conversion ratio, kg/kg	1.916	1.950	1.915	0.040	0.786

^1^ CON (control), basal diet; CV, CON + 0.5% *Chlorella vulgaris*; TO, CON + 0.5% *Tetradesmus obliquus*. ^2^ SEM, standard error of the means.

**Table 4 animals-13-01589-t004:** Effect of dietary microalgae on egg quality in laying hens.

Item	CON ^1^	CV	TO	SEM ^2^	*p*-Value
Yolk color	7.49 ^b^	8.14 ^a^	7.94 ^a^	0.108	0.001
Haugh unit	93.73	94.63	95.73	0.899	0.308
Eggshell strength, kg/cm^2^	5.595	5.462	5.389	0.183	0.723
Eggshell thickness, mm	0.418	0.422	0.415	0.005	0.619
Eggshell color, unit	23.78	24.28	23.61	0.714	0.790

^a,b^ Mean values without a common superscript within the same row differ (*p* < 0.05). ^1^ CON (control), basal diet; CV, CON + 0.5% *Chlorella vulgaris*; TO, CON + 0.5% *Tetradesmus obliquus*. ^2^ SEM, standard error of the means.

**Table 5 animals-13-01589-t005:** Effect of dietary microalgae on jejunal histology in laying hens.

Item	CON ^1^	CV	TO	SEM ^2^	*p*-Value
Villus height, μm	585.9	583.3	568.3	23.19	0.847
Crypt depth, μm	168.4	175.4	176.9	8.815	0.773
Villus height to crypt depth ratio	3.58	3.35	3.22	0.166	0.323

^1^ CON (control), basal diet; CV, CON + 0.5% *Chlorella vulgaris*; TO, CON + 0.5% *Tetradesmus obliquus*. ^2^ SEM, standard error of the means.

**Table 6 animals-13-01589-t006:** Effect of dietary microalgae on the percentages of cecal short-chain fatty acids in laying hens.

Item	CON ^1^	CV	TO	SEM ^2^	*p*-Value
Acetate	66.75	65.68	64.57	0.686	0.137
Propionate	15.21	15.93	15.36	0.745	0.787
Isobutyrate	1.273	1.553	1.355	0.230	0.694
Butyrate	12.87	12.34	14.01	1.029	0.542
Isovalerate	1.28	1.77	1.76	0.195	0.186
Valerate	2.61	2.74	2.94	0.190	0.516

^1^ CON (control), basal diet; CV, CON + 0.5% *Chlorella vulgaris*; TO, CON + 0.5% *Tetradesmus obliquus*. ^2^ SEM, standard error of the means.

**Table 7 animals-13-01589-t007:** Effect of dietary microalgae on antioxidant and immune markers in ileal mucosa of laying hens.

Item	CON ^1^	CV	TO	SEM ^2^	*p*-Value
Superoxide dismutase, U/mg protein	1.496	1.577	1.516	0.128	0.896
Malondialdehyde, nmol/mg protein	1.73	1.83	1.30	0.265	0.343
Catalase, U/mg protein	34.9	36.3	29.4	2.579	0.167
Glutathione peroxidase, U/mg protein	0.074	0.089	0.076	0.010	0.566
Secretory immunoglobulin A, μg/mg protein	38.6	38.9	44.1	2.637	0.277

^1^ CON (control), basal diet; CV, CON + 0.5% *Chlorella vulgaris*; TO, CON + 0.5% *Tetradesmus obliquus*. ^2^ SEM, standard error of the means.

## Data Availability

Not applicable.

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
