# Peer review of "Effect of Dietary Chlorella vulgaris or Tetradesmus obliquus on Laying Performance and Intestinal Immune Cell Parameters"

_animals, 2023, doi:10.3390/ani13101589_

Round 1

Reviewer 1 Report

All comments are in the text.

Author Response

Line 210-212: If there was no difference, it does not make sense speak about highest value.

Authors’ Response: Per the Reviewer’s comment, we have now deleted the statment from the text.

Line 214: the same to this phrase.

Authors’ Response: Per the Reviewer’s comment, we have now corrected as follows (Lines 219-221).

“Dietary CV tended to increase the subpopulation of CD4+ T cells compared with the control and the TO-fed groups, but statistical significance was not found (P = 0.054).”

In addition, we have indicated that tendency was declared at P < 0.10 in Statistical Analysis section (L180-181).

Figure 2. (B) TCR γδ cells (CD8-) : Is it missing the probability here?

Authors’ Response: Per the Reviewer’s comment, we add p-value of CD8- γδ T cells. Thank you for kind comment (Figure 2).

In addition to the comments above, all minor comments written in the pdf file have been corrected in the revised manuscript.

Reviewer 2 Report

There is a major issue with this work, and that is that the authors are not reporting their results correctly, and therefore, the conclusions that emanate from it are not correct.

If we decide to use statistical analysis to validate our results, then we must follow it.

The comparison of CD4+ T cell subpopulations shows no significant difference between treatments. The p-value of the comparison is 0.054, which is not significant. With that, the whole story changes, as there is no change in favor of using microalgae in diets of laying hens. The authors themselves report, as usual, that the level of significance was set at p<0.05.

Therefore, the article should be rewritten reporting appropriately the results and their scope.

Author Response

There is a major issue with this work, and that is that the authors are not reporting their results correctly, and therefore, the conclusions that emanate from it are not correct.

If we decide to use statistical analysis to validate our results, then we must follow it.

The comparison of CD4+ T cell subpopulations shows no significant difference between treatments. The p-value of the comparison is 0.054, which is not significant. With that, the whole story changes, as there is no change in favor of using microalgae in diets of laying hens. The authors themselves report, as usual, that the level of significance was set at p<0.05.

Therefore, the article should be rewritten reporting appropriately the results and their scope.

Authors’ Response: We really appreciate the reviewer’s critical comments. In this study, we did our best not to over-exaggerate the findings. We have now added a statement that tendency was declared at p < 0.10 in Statistical Analysis section. As explain in our response to Reviewer #1, we have deleted the non-significant effect from the text. In addition, we have expanded our discussion per the immune-related questions raised by the Reviewer #3.

Reviewer 3 Report

It is very well-written paper and the authors have done a good job of describing all the results and tried very well to discuss those findings. However, my major criticism/comments as below:

1. The treatments had no much 'significant' effect on the performance parameters and others EXCEPT the immune parameters. In such a case, I would like the authors to give it an extended thought as to how best they can talk a little more about their findings. For example,

Why CD4+ T cell frequencies matter in the gut in relation to immune effector Vs regulatory phenotypes and how their findings may suggest a immunomodulatory role?

Importance of gdT cells in the gut? What's the role of CD8-ve Vs CD8+ve cells? Are these IELs? Why immunomodulation via gdT cells is important in the gut immune homestasis?

2. Authors should analyze DP cells (CD3+CD4+CD8+) as these cells are important in regulation too (in chickens).

3. Bu-1 is a weak stainer and hard to conclude on B cells just based on this antibody. SHould have used IgM to gate B cells. Anyways, authros should mention this in results.

4. A thorough explanation of 'gating strategy' SHOULD be included in the materials methods section. Did you use FMO gating control? How did you remove doublets etc? Pl note that the CD4 and Bu-1 numbers are pretty low and one would definitely like to see what were your gating control numbers? 0 or less than 0.1-0.5%? Maybe a good idea you add a 'supplementary figure' to show your gating strategy?- Not mandatory but optional.

5. The title " Dietary Chlorella vulgaris or Tetradesmus obliquus fails to affect laying performance, but modulates gut immune parameters 3 in laying hens" . My objection is to the highlighted terms. Fails is very assertive but actually, authors didn't test different % of dietary inclusion. These algae may be useful but just didn't show benefits under the conditions they used. 'modulates' is a a BIG assumption when authors haven't really checked many cellular and molecular mechanisms of regulation. I would suggest a change to "Effect of dietary Chlorella vulgaris or Tetradesmus obliquus on the laying hen performance and intestinal immune cell parameters".

Minor comments:

Line 71: Since this was an immunological study, you should state if the birds were vaccinated.

Line 96: “...consecutive days at 4 week” week should be ‘weeks’.

Line 127: Add where Bioassay Systems’s headquarters is located in the USA

Line 187-188: You do not need to add (p > 0.05) if something is not significant, just state it is not significant.

Line 284-287: You are saying that dietary microalgae could potentially enhance immune development and competence in facilitating productive immune responses against pathogenic bacteria and parasites, however there is no real evidence to support this from your paper besides the change in γδ TCR in one of the algae tested. Potentially changing the word “would” on line 285 to ‘could’ would allow you to change this from a concrete statement, to a potential estimate that needs to be further tested.

Line 295: You don’t need to describe what less optimal environments are “ (i.e., stocking density, heat stress)” in the concluding paragraph. If you wish to do that it would benefit from being placed in the introduction or the discussion section instead.

In the references section, algae species should be italicized in the titles, for example “Tetradesmus obliquus” in line 335 and 338, and so on, should all be italicized. Additionally other species, such as Eimeria in line 355 need to be italicized.

Author Response

It is very well-written paper and the authors have done a good job of describing all the results and tried very well to discuss those findings. However, my major criticism/comments as below:

1. The treatments had no much 'significant' effect on the performance parameters and others EXCEPT the immune parameters. In such a case, I would like the authors to give it an extended thought as to how best they can talk a little more about their findings. For example, Why CD4+ T cell frequencies matter in the gut in relation to immune effector Vs regulatory phenotypes and how their findings may suggest a immunomodulatory role? Importance of gdT cells in the gut? What's the role of CD8-ve Vs CD8+ve cells? Are these IELs? Why immunomodulation via gdT cells is important in the gut immune homestasis?

Authors’ Response: We appreciate the Reviewer’s critical comments. In this study, the analysis of immune cells in the lamina propria layer in the laying hen small intestine was conducted, and we found that both CD4 T cells and γδ T cells were altered by the microalgae feed. Per the Reviewer’s comments, we have now expanded our discussion on the role of immune cell subpopulations in chickens that are relevant to our findings as follows:

“However, B cells and monocytes/macrophages, and T cell subpopulations (i.e., CD8+ T cells and CD8+ TCR γδ T cells) were not altered by dietary microalgae. Among the T cells, CD4+ T cells play a key role in the gut homeostasis. For example, it is known that they stimulated the phagocytosis of antigen presenting cells (APCs) or regulated other immune cells through the cytokine secretion [21]. There is also supporting evidence for the existence of CD4+ T cell subsets, such as Th1, Th2, Th17, and Treg cells [22,23]. To identify the characteristics of laying hen CD4+ T cells and subsets, the related genes or proteins expression analysis should be applied in future studies. Another major finding of this study is that TCR γδ T cells (CD8-) were lower in the TO feed group compared with control diet group. There are two major chicken T cells populations, αβ- and γδ- TCR T cells [24]. Unlike other mammals, the ratio of γδ T cells in chickens is high (i.e., blood circulating 20%-50%, mucosal tissue up to 50%), and they probably play an important role in the chicken’s immune systems, including gut immunity [25]. Diverse functions of chicken γδ T cells have been reported in the previous studies, such as production of interferon gamma (IFN- γ) and cytotoxic activity [26,27]. The cytotoxic function of chicken γδ T cells (especially CD8+ γδ T cells) has been reported in Marek’s disease virus challenge study [28]. Briefly, TCR γδ stimulation by viral infection induced IFN-γ production by γδ T cells in chicken peripheral blood mononuclear cells (PBMCs), and the injection of activated PMBCs reduced viral replication, and induced cytokine secretion and cytotoxic activity to regulate viral infection. Fenzl et al. reported that CD8+ γδ T cells had higher cytotoxic activity compared to CD8- γδ T cells [27]. In our finding, there was a significant reduction in CD8- γδ T cells, not CD8+ γδ T cells, in the laying hen small intestine LP according to the microalgae TO feed. Although studies related to CD8- γδ T cells is insufficient compared to CD8+ γδ T cells, changes in CD8- γδ T cells by dietary feed additives will play an important role in gut immunity (especially cytotoxicity function). In future studies, Ex vivo or in vivo experiments should be conducted to identify the characteristics of chicken γδ T cells.”

2. Authors should analyze DP cells (CD3+CD4+CD8+) as these cells are important in regulation too (in chickens).

Authors’ Response: We have analyzed the double positive T cells and presented them in Figure 2B. Unfortunately, there is no significant difference between groups.

3. Bu-1 is a weak stainer and hard to conclude on B cells just based on this antibody. SHould have used IgM to gate B cells. Anyways, authros should mention this in results.

Authors’ Response: Thank you for valuable comment. We used only Bu-1 antibody to identify B cell population of laying hen small intestine lamina propria cells. The Bu-1 antigen is expressed by chicken B cells throughout most of their development. It is also well known as a strong B cell marker and has been widely used in many chicken studies, including laying hens. As the Reviewer critically pointed out, it is important that additional antibodies such as IgM will be needed to make more precise conclusions. Unfortunately, it was not available for this study, and it may be used for the future studies. In the revised Results section, we have added information of immune cell markers to clearly communicate our result to readers.

4. A thorough explanation of 'gating strategy' SHOULD be included in the materials methods section. Did you use FMO gating control? How did you remove doublets etc? Pl note that the CD4 and Bu-1 numbers are pretty low and one would definitely like to see what were your gating control numbers? 0 or less than 0.1-0.5%? Maybe a good idea you add a 'supplementary figure' to show your gating strategy?- Not mandatory but optional.

Authors’ Response: Unfortunately, fluorescence minus one (FMO) control did not include in this study. On the other hand, the antibodies used in this study are widely used clone, and are also believed to be reliable. And flow cytometry staining clearly separated cell subpopulations in this study. We also removed doublet cells through FSC-A and FSC-H in the flowjo1 software. Per the Reviewer’s recommendation, we have now added the gating strategy of the two panels as supplementary Figure 1. In addition, we showed the percentage of the Bu-1+ cell among MHCII + cells and CD4+ cells among CD3+ T cells.

5. The title " Dietary Chlorella vulgaris or Tetradesmus obliquus failsto affect laying performance, but modulatesgut immune parameters 3 in laying hens" . My objection is to the highlighted terms. Fails is very assertive but actually, authors didn't test different % of dietary inclusion. These algae may be useful but just didn't show benefits under the conditions they used. 'modulates' is a a BIG assumption when authors haven't really checked many cellular and molecular mechanisms of regulation. I would suggest a change to "Effect of dietary Chlorella vulgaris or Tetradesmus obliquus on the laying hen performance and intestinal immune cell parameters".

Authors’ Response: Per the Reviewer’s comment, we have now corrected the title.

“Effect of dietary Chlorella vulgaris or Tetradesmus obliquus on the laying performance and intestinal immune cell parameters”

Minor comments:

Line 71: Since this was an immunological study, you should state if the birds were vaccinated.

Authors’ Response: Per the Reviewer’s comment, we have now added the vaccination program in the text as follows.

These hens had been vaccinated against a variety of diseases including Marek’s disease, Newcastle disease, infectious bronchitis, and infectious bursal disease, but no vaccines were practiced after 12 weeks of age.

Line 96: “...consecutive days at 4 week” week should be ‘weeks’.

Authors’ Response: Per the Reviewer’s comment, we have now corrected (Line 95).

“On the last three consecutive days at 4 weeks”

Line 127: Add where Bioassay Systems’s headquarters is located in the USA

Authors’ Response: Per the Reviewer’s comment, we have now corrected (Lines 125-126).

“Both the kits were bought from BioAssay Systems (Hayward, CA, USA).”

Line 187-188: You do not need to add (p > 0.05) if something is not significant, just state it is not significant.

Authors’ Response: Per the Reviewer’s comment, we have now deleted.

Line 284-287: You are saying that dietary microalgae could potentially enhance immune development and competence in facilitating productive immune responses against pathogenic bacteria and parasites, however there is no real evidence to support this from your paper besides the change in γδ TCR in one of the algae tested. Potentially changing the word “would” on line 285 to ‘could’ would allow you to change this from a concrete statement, to a potential estimate that needs to be further tested.

Authors’ Response: Per the Reviewer’s comment, we have now corrected (Lines 316-318).

“Thus, dietary microalgae (CV and TO) could have their potential enhancing immune development and competence in the intestine facilitating protective immune response against pathogenic bacteria and parasites in laying hens.”

Line 295: You don’t need to describe what less optimal environments are “ (i.e., stocking density, heat stress)” in the concluding paragraph. If you wish to do that it would benefit from being placed in the introduction or the discussion section instead.

Authors’ Response: Per the Reviewer’s comment, we have now corrected (Lines 324-326).

“Further research is required to determine whether dietary microalgae would augment host immune competence under pathogen exposure in chickens.”

In the references section, algae species should be italicized in the titles, for example “Tetradesmus obliquus” in line 335 and 338, and so on, should all be italicized. Additionally other species, such as Eimeria in line 355 need to be italicized.

Authors’ Response: Per the Reviewer’s comment, we have now corrected the references.

Round 2

Reviewer 2 Report

The authors made the main modifications suggested and refined their conclusions based on their results.

Author Response

We appreciate the Reviewer's comment.

Reviewer 3 Report

Revised version reads good and satisfactorily addressed all my comments.

Author Response

We appreciate the Reviewer's comment.